# Effect of Insoluble Dietary Fiber Extracted from Feijoa (*Acca sellowiana* (O. Berg) Burret.) Supplementation on Physicochemical and Functional Properties of Wheat Bread

**DOI:** 10.3390/foods12102019

**Published:** 2023-05-16

**Authors:** Dan Wang, Qingming Wang, Yunfei Sun, Zilong Qing, Junhui Zhang, Qiyang Chen

**Affiliations:** 1School of Life Science and Engineering, Southwest University of Science and Technology, Mianyang 621010, China; wanda@swust.edu.cn (D.W.); 18881626632@163.com (Y.S.); qzl2782448021@163.com (Z.Q.); zjh1234560228@163.com (J.Z.); 2Lu’an Academy of Agricultural Sciences, Lu’an 237001, China; w332686960@163.com

**Keywords:** wheat bread, feijoa fruit, insoluble dietary fiber, physicochemical properties, functional properties

## Abstract

This study aimed to assess the effects of insoluble dietary fiber (IDF) from feijoa supplementation on the physicochemical and functional properties of wheat bread. The results showed that feijoa IDF (FJI) had the typical structures of hydrolysis fiber, polysaccharide functional groups, and crystal structure of cellulose. The gradual increase of FJI levels (from 2 to 8%) in wheat bread resulted in increased total DF, ash, and protein contents, accompanied by a reduction in moisture, carbohydrates, and energy value. The inclusion of FJI in the bread crumb caused a rise in both redness (*a**) and yellowness (*b**) values while decreasing the brightness (*L**) relative to the control specimen. In addition, adding FJI up to 2% significantly increased total phenolic and flavonoid contents and antioxidant activity, as well as flavor score of supplemented bread samples, while additions above 2% resulted in undesirable taste and texture. FJI addition caused higher bile acid, NO^2−^, and cholesterol adsorption capacities. Moreover, FJI addition up to 4% significantly reduced glucose adsorption capacities at different in vitro starch digestion intervals. The findings revealed that FJI offers great potential as an ideal functional ingredient in food processing.

## 1. Introduction

Wheat flour bread is one of the critical staples in many countries, which contains high carbohydrates but has low functional ingredients and fewer health benefits [1]. Wheat flour has lost a significant amount of dietary fiber (DF), vitamins, and other nutrients over the years due to the growth of the flour milling industry, nutritional factors, and economic factors [2]. Consumers generally prefer foods with high protein and fiber content to maintain their health and act against many diseases, such as diabetes and obesity. As a result, a contemporary market trend is emerging with the development of products that offer both health benefits and desirable sensory attributes [3]. The nutritional value of bread formulations can be enhanced by incorporating various ingredients, including DF, folic acid, vitamin E, and polysaccharides [4]. Adding DF from fruit and vegetable sources is the most effective way to improve the nutritional values of bread [5].

Feijoa (*Acca sellowiana*) is a woody species cultivated in some regions of the world, such as New Zealand, the USA, France, China, and Australia. Researchers at Shanghai Botanical Garden first brought feijoa to China from Europe in the 1980s. After that, it was introduced to Jiangsu, Sichuan, and Hunan provinces [6]. Feijoa fruit is sweet/sour and highly aromatic, which has gained global attention because of its potentially beneficial health properties and delicious eating quality [7,8]. The edible fruit ripens in autumn as a spherical berry and contains abundant bioactive compounds, such as phenolics and carotenoids [9]. In addition to fresh consumption, fruits can be processed and incorporated into producing various food and beverage items. Market research reveals that a range of feijoa-derived products, including ice cream, chocolate, candy, wine, jam, yogurt, and muffins, has been developed. Feijoa-based ingredients have the potential to be developed for functional food formulation [10]. As an essential source of DF, feijoa fresh fruit contains 6.4 g of DF per 100 g [8]. A large amount of insoluble dietary fiber (IDF) is often discarded during food processing, resulting in waste and polluting the environment.

The promotion of DF on health has been widely documented. Increased DF intake has decreased the risk of certain chronic diseases, such as diabetes, obesity, and cardiovascular disease [11]. DF can be divided into water-soluble (SDF) and water-insoluble (IDF). IDF regulates intestinal function by improving peristalsis. A daily dietary intake of 50–75% IDF is recommended to obtain the maximum benefits of DF [4]. In addition, IDF affects oil and water holding capacity, which can be used to develop new foods or reformulate existing ones [12]. Several studies have been published on the effect of fruit IDF on wheat bread and cookies [4,13]. The supplementation of DF significantly influences the physicochemical, functional, and sensory properties of wheat bread, which affects consumer choice [14,15]. However, no reports were found on the use of IDF from feijoa residue as an ingredient in wheat bread formulations. This study aimed to examine the effects of feijoa IDF (FJI) supplementation on the physicochemical and functional properties of wheat bread.

## 2. Materials and Methods

### 2.1. Materials and Chemicals

Fresh feijoa (*Acca sellowiana* (O. Berg) Burret.) was collected from the back mountain farm of Southwest University of Science and Technology (Mianyang, China) in October 2021. High-gluten flour, butter, sugar, non-fat dry milk, salt, and dry yeast were purchased from a local supermarket. The major reagents including 1,1-Diphenyl-2-picrylhydrazyl radical (DPPH), 2,2′-azinobis 3-ethylbenzothiazoline-6-sulfonic acid (ABTS), 6-hydroxy-2,5,7,8-tetramethylchroman-2-carboxylic acid (Trolox), and 2,4,6-tris (2-pyridyl)-S-triazine (TPTZ) were obtained from Sigma-Aldrich (St. Louis, MO, USA). The remaining chemicals and solvents were purchased from Chuandong Chemical Reagent Co., Ltd. (Chongqing, China).

### 2.2. Preparation of FJI

FJI was prepared according to the reported methods of Hua, Lu, Qu, Liu, Zhang, Li, Chen, and Sun [12] and Bunzel et al. [16]. Briefly, feijoa pulps were boiled to remove polysaccharides, and the residue was washed with ethanol and distilled water to remove water-soluble infractions. Then, the air-dried pulp was ground into powder and sieved through the 60-mesh. The feijoa powder (120 g) was mixed with 3.6 mL heat-stable α-amylase in 3.0 L of buffer (pH 6.0) and incubated at 95 °C for 20 min. Subsequently, the pH of reactants was adjusted to 7.5, and the protease (6.0 mL) was added and placed into a water bath for 1 h at 60 °C. Then, amyloglucosidase (2.4 mL) was added, and the pH was adjusted to 4.5 at 60 °C to remove starches and proteins. Finally, after centrifugation at 5000 rpm for 10 min, the solid precipitate was washed with 95% ethanol and distilled water. The insoluble fraction was precipitated by centrifugation (5000 rpm, 10 min) and then air-dried (60 °C for 48 h) to achieve a moisture content below 6%, and the feijoa IDF was obtained. The dried feijoa IDF was ground using a disc mill (FZ102, Tester Instruments Ltd., Tianjin, China) and sieved at # 200 mesh. The feijoa IDF was packaged in a self-sealing bag and stored at room temperature until further analysis.

### 2.3. FJI Structure Analysis

EVO 18 Scanning Electron Microscope (Carl Zeiss company, Oberkochen, Germany) was used to observe the microscopic appearance of feijoa IDF. The scanning images were captured at accelerating voltages of 25 kV and photographed at magnifications of 5000× (scale bar 10 μm).

Particle size distribution of feijoa IDF was estimated using a Matersizer 2000 Laser Particle Sizer (Malvin Instruments Ltd., Worcestershire, UK). Particle size distribution parameters were measured as different diameters, such as D_10_ (μm), D_50_ (μm), and D_90_ (μm). The span value was calculated using the Span = (D_90_ − D_10_)/D_50_.

Fourier transform infrared spectroscopy (FT-IR) analyses were performed using a PerkinElmer Spectrum One spectrophotometer (PerkinElmer, Waltham, MA, USA) equipped with universal attenuated total reflectance (ATR). The spectra were read in the range of 4000 cm^−1^ to 400 cm^−1^ with a resolution of 4 cm^−1^ and 32 scans per sample.

X-ray diffractometry (XRD) patterns were obtained by using an X-ray diffractometer (X’Pert PRO, Panaco Co., Almelo, The Netherlands) with Cu-Kα radiation source (λ = 1.542 Å, 50 mA, 50 kV) in the diffraction angle (2θ) of 3–80°.

### 2.4. Hydration Properties of FJI

Water holding capacity (WHC) and oil holding capacity (OHC) were measured according to the method of Raghavendra et al. [17] with minor modifications. Briefly, a sample (0.5 g) was added into 10 mL of distilled water (for WHC) or peanut oil (for OHC) and hydrated at 25 °C for 24 h. After centrifuging at 4000× *g* for 10 min, the supernatant was discarded. The results were expressed as grams of water/oil held by 1 g of sample in dry weight (DW).

Swelling capacity (SC) was evaluated according to Raghavendra, Rastogi, Raghavarao, and Tharanathan [17]. The sample (0.2 g) was mixed with distilled water and rested at 25 °C for 18 h. The SC was analyzed according to the equation:$$\mathrm{SC}\;\left(\mathrm{mL}/g\right)=\frac{\left(V_{1}-{\;V}_{0}\right)}{W_{0}}$$
where V_0_ and V_1_ were the volume before and after swelling, respectively; W_0_ was the sample weight.

### 2.5. Determination of Mineral Content of FJI

Mineral content was determined according to a method reported by Hua, Lu, Qu, Liu, Zhang, Li, Chen, and Sun [12]. Feijoa IDF (0.5 g) was added with 15 mL HNO_3_/HClO_4_ (4:1) and heated on an electric hot plate in a beaker until the solution became apparent and its volume decreased to nearly 1 mL. After digestion, the solution was diluted with ultra-pure water to the final volume of 50 mL in a volumetric flask. In order to determine the concentration of K, Na, and Mn elements, the test solution was subjected to respective dilutions of 200×, 6×, and 20×. An Analyst 7000 atomic absorption spectrometer (Shimadzu Co., Kyoto, Japan) equipped with flame atomic absorption spectroscopy (FAAS for Ca, Fe, K, Na, and Zn) and graphite furnace (GFAAS for Cu and Mn) was used to analyze the elemental composition.

### 2.6. Bread-Making Process

The bread was prepared using the scalding process and secondary fermentation method. Firstly, 120 g boiling water was mixed with 100 g wheat flour (protein content 12.2 g/100 g) to create hot dough. The bread formulation included 194 g of wheat flour, 12.5 g of hot dough, 116 g of water, 8.5 g of non-fat dry milk, 30.0 g of butter (fat content 81.0 g/100 g), 25.0 g of sugar, 3.0 g of dry yeast, and 2.50 g of salt. Wheat flour was replaced with FJI at 0.0, 4.0, 8.0, and 16.0 g/200 g flour. According to the bread formulation, the hot dough, wheat flour, FJI, and other dry ingredients were blended using a Sanlida mixer (1590, Sanlida Co., Ltd., Shenzhen, China) for 30 min; then, butter was added and was mixed for another 15 min. For the primary fermentation, the dough was placed in a thermostat at 28 °C and 75% relative humidity (RH) for 60 min. The fermented dough was divided into three equal parts and proofed at room temperature for 20 min before being exhausted. Following fermentation at 38 °C and 75% RH for 60 min (secondary fermentation), the bread doughs were baked at 190 °C in a baking oven (SEC-4Y, Sain-Mate, Ma-chinery Co., Ltd., Guangzhou China) for 25 min, followed by cooling (2 h) at room temperature (25 °C). Each recipe made three loaves.

### 2.7. Determination of Proximate Composition

The bread was measured using the methods recommended by AOAC [18] and AOAC [19] to determine its approximate composition (moisture, total DF, fat, ash, and protein). The carbohydrate content of the sample was determined by subtracting protein, moisture, fat, and ash from its total weight. A formula was used to calculate the energy value of bread (kJ/100 g):$$\mathrm{Caloric}\;\mathrm{value}\;\left(\frac{\mathrm{kJ}}{100g}\right)=\left(g\;\mathrm{carbohydrates}\;\times 4+\;g\;\mathrm{fat}\;\times 9+\;g\;\mathrm{protein}\;\times 4\right)\times 4.184$$

### 2.8. Determination of the Specific Volume

The loaf volume was determined using a standard millet displacement method. The AACC Approved Method no. 10-05.01 [20] was used to measure the volume of the bread (mL). Bread weight (g) was within 0.01 g of accuracy. Volume/weight (mL/g) was used to calculate bread-specific volume. Each sample was measured three times.

### 2.9. Color Analysis

The crust and crumb colors of bread were measured using a colorimeter (CM-3600 A, Konica Minolta, Osaka, Japan) using Hunter’s scale for *L**, *a**, and *b** at room temperature, where *L** denotes the color brightness, *a** denotes red/green saturation, and *b** denotes yellow/blue saturation. Crust color was evaluated at 3 points on each surface of the bread. Crumb color was determined at the middle point of the central 2 cm thick slice.

### 2.10. Texture Analysis

According to the report of Jiang, Feng, Wu, Li, Bai, Zhao, and Ameer [4], the texture profile analysis (TPA) was performed using a texture analyzer (TA-XT Plus, Stable Micro Systems, UK). A 1 cm thick slice of bread was cut from the loaf. Bread slice cores were subjected to texture analysis using a sample of 1 cm × 2 cm. Slices of bread were compressed to 50% deformation using a cylindrical probe P/36R. During the test, the speed was set to 1 mm/s; the interval between the first and second compressions was 5 s, and the trigger force was 5 g. Several characteristics were evaluated, including hardness, chewiness, springiness, and resilience.

### 2.11. Determination of TPC, TFC, and Antioxidant Activity

Samples were extracted based on our previous report [21], with minor modifications. Briefly, the bread sample (1.0 g) or feijoa IDF (0.4 g) was thoroughly mixed with 80% methanol (10 mL), ultrasonically extracted for 30 min at room temperature using 300 W, and then centrifuged at 5000 rpm for 10 min. The supernatants from the three extractions were collected, and methanol was added to reach a final volume of 50 mL. The TPC was attained following the Folin–Ciocalteu method and was expressed in mg gallic acid equivalent (mg GAE/100 g dry weight (DW)) [21]. The Al(NO_3_)-NaNO_2_ complexometric method was used to analyze bread samples for their TFC as per Dong et al. [22]. TFC was expressed as mg rutin equivalent (RE)/100 g DW. ABTS, DPPH, and FRAP analysis were performed according to our previously reported method [21]. ABTS, DPPH, and FRAP values were determined at the wavelength of 734, 515, and 593 nm, respectively, using a spectrophotometer (LAMBDA 25 UV/Vis spectrophotometer, PerkinElmer, Waltham, MA, USA). These results were expressed as micromolar Trolox equivalents (μmol TE/g, DW).

### 2.12. Determination of Adsorption Capacities

#### 2.12.1. Bile Acid-Adsorption Capacity

Based on the conditions described by Zheng et al. [23] with slight modifications, the bile acid-adsorption capacity of the sample was determined. Specifically, the sample (1 g) was dissolved in sodium cholate (30 mL, 3 mg/mL, pH 7.0), dispersed at 180 r/min, and shaken at a constant 37 °C for 2 h. A solution of 0.5 mL supernatant was combined with 0.5 mL furfural (0.3%, *w/v*) and 3 mL sulfuric acid (45%, *w/v*) and reacted for 30 min. An absorbance measurement at 620 nm was conducted. An analysis of sodium cholate residuals was performed to determine bile salt binding ability.

#### 2.12.2. Nitrite-Adsorption Capacity

The nitrite adsorption capacity was carried out as described by Hua, Lu, Qu, Liu, Zhang, Li, Chen, and Sun [12] by mixing a sample (1 g) with sodium nitrite solution (25 mL, 100 μmol/L), followed adjustment of pH to 2.0 or 7.0 and incubation at 37 °C for 2 h in a constant temperature shaker (120 rpm). A total of 500 μL of supernatant was mixed with acetic acid (2.5 mL, 60%), N-(1-Naphthyl) ethylenediamine dihydro-chloride (2.5 mL, 1 g/L), and sulfanilic acid (2.5 mL, 10 g/L) and incubated for 25 min in the dark. The absorbance of the supernatant was determined at 538 nm.

#### 2.12.3. Cholesterol-Adsorption Capacity

The cholesterol adsorption capacity was determined according to Hua, Lu, Qu, Liu, Zhang, Li, Chen, and Sun [12]. Briefly, the bread sample (1 g) was fully mixed with 30 mL of diluted egg yolk emulsion (9 times distilled water dissolved in fresh egg yolk). The mixture was adjusted to pH 2.0 and 7.0, respectively, and incubated for 180 min at 37 °C. The mixture was centrifuged at 4000× *g* for 15 min. The absorbance of the supernatant was determined at 550 nm. The absorbance of egg emulsion cholesterol was blank, and the cholesterol adsorption capacity was expressed as mg/g.

### 2.13. In Vitro Starch Digestion

The in vitro starch digestion assay was performed as described by Jiang, Feng, Wu, Li, Bai, Zhao, and Ameer [4]. Bread samples (0.5 g) were placed in 50 mL screw-capped plastic tubes, and 1 mL of α-amylase (250 U/mL, 0.2 M pH = 7 carbonate buffer) was added, and then 5 mL of pepsin suspension (Sigma P-6887; 1 mL/mL 0.02 M pH = 2 HCl) was added after 15-20 s. The mixture was incubated at 37 °C for 30 min and neutralized with NaOH (5 mL, 0.02 M), and sodium acetate buffer (25 mL, 0.2 M, pH = 6) and 5 mL of trypsin/amylase mixture were successively added. Aliquots (1 mL) were taken and mixed with 2 mL of anhydrous ethanol after 0, 20, 60, 90, and 120 min to stop the enzyme reaction. A 3.5-dinitrosali-cylic acid (DNS) method [24] was used to detect glucose release. In brief, reaction liquid (0.5 mL) was added to 1.5 mL of distilled water and 1.5 mL of DNS. Then, the mixture was boiled for 5 min, after cooling to room temperature, and diluted to 20 mL with distilled water. The absorbance was measured at 540 nm. Reducing sugar concentration in each sample was expressed as glucose equivalents based on a glucose calibration curve.

### 2.14. Sensory Evaluation

The sensory evaluation was conducted by 20 untrained students from the School of Life Sciences and Engineering, College of Food, Southwest University of Science and Technology. All participants signed informed consent (Supplementary material), and this study does not involve ethics or moral issues. The sensory characteristics (appearance, flavor, taste, texture, and overall acceptability) were evaluated using a 9-point hedonic scale. Based on the ranking system, point 1 represented extreme dislike, point 5 denoted neither like nor dislike, and 9 indicated extreme like.

### 2.15. Statistical Analysis

All experimental measurements were performed in triplicate (*n* = 3). The recorded data are expressed as mean ± standard deviation (SD). Statistical analyses were performed using SPSS version 18.0 (Chicago, IL, USA), and variance was analyzed by one-way analysis of variance (ANOVA). Significant differences (*p* ≤ 0.05) between the mean values were determined using Duncan’s Multiple Range Test.

## 3. Results and Discussion

### 3.1. Mineral Composition and Physicochemical Properties of FJI

As shown in Table 1, the mineral element results presented that FJI contained abundant mineral elements the human body needs. Seven mineral elements were observed in FJI with a total content of 3.72 mg/g. K was the predominant element, with a content of 2847.47 μg/g, similar to the results of previous studies, and a quarter of the peel flour content [9], which suggested that a high level of potassium might be retained in IDF during processing. FJI had a high concentration of Zn (80.89 mg/kg), which was obviously higher than Ginseng-IDF [4]. Except for its nutritional value, FJI had desirable physicochemical properties. The WHC of FJI was 4.66 g/g, lower than that of Ginseng-IDF (17.66 g/g) but much higher than that of the bamboo shoot shell IDF (2.83 g/g) [25]. The OHC index of FJI (4.18 mg/g) was higher than that of ginseng-IDF (1.78 g/g) [4]. In addition, the SC level of FJI (2.09 mL/g) was close to that of rice bran DF [26]. The level of antioxidant activity was closely related to the TPC. FJI showed relatively low TPC, TFC, and antioxidant activities, indicating that the TPC of feijoa fruit was mainly present in the soluble fraction. FJI was superior to bamboo shoot shell IDF in bile acid salt and cholesterol adsorption, while the opposite was true for nitrite ion (NO^2−^) adsorption [25]. The effect of IDF’s hydration and adsorption properties was shown to depend on the cellulosic material’s processing and structural characteristics [27], which may help explain why enzymatically treated FJI could match the OHC level of superfine rice bran IDF.

### 3.2. Structure Analysis of FJI

The microstructure of FJI powder was presented in Figure 1A. The surface of FJI was rough, with few voids and irregular fragments. The D_50_ value of the fine FJI was 66.15 μm, and the specific surface area was 96.56 m^2^/kg (Figure 1B). Fine particles suggested a great number of particles per unit weight, high dispersibility, and high solubility in food systems [28]. Qualitative analysis of polysaccharide functional groups was shown as an FTIR spectrograph in Figure 1C. The prominent peaks at 3426 cm^−1^ in the FJI have been assigned to the O–H stretching vibration by molecular hydrogen bonding of uronic acid [29]. The peak at 2922 cm^−1^ corresponds to the C–H stretching of –CH_3_ or = CH_2_, which indicated the typical structure of cellulose polysaccharide compounds [12]. The minor peak at 1743 cm^−1^ resulted from the acetyl group or C=O bonds of hemicellulose [29]. The notable peak at 1640 cm^−1^ was mainly due to esterified and ionized carboxyl groups of GalUA [30]. The weak peak at 1514 cm^−1^ was mostly attributed to the aromatic or aliphatic C–H group vibration of lignin [12]. The weak peak at 1379 cm^−1^ was a result of vibration specific for cellulose or hemicellulose [31]. The weak peak at 1259 cm^−1^ was mainly from the O–H or C–O vibration of hemicellulose [12]. The peaks at 1048 cm^−1^ could be ascribed to C–O stretching vibration of the guaiacyl unit of lignin or the CO–OR stretching vibration of hemicelluloses [12]. These results indicated that FJI had the typical functional groups of cellulosic polysaccharides, including pectin, cellulose, hemicellulose, and lignin.

The results of X-ray diffractometry revealed that FJI possessed a cellulose I crystal structure with two prominent diffraction peaks of 2θ = 16.14° and 21.59° (Figure 1D), suggesting the presence of a crystalline cellulose region. The result was comparable to those observed in rice bran-IDF and ginger-IDF [12,32], where partial destruction of the amorphous fraction of cellulose was detected. The irregular weak peak at 2θ = 34.49° could be attributed to the denaturation of cellulose during the FJI extraction process under conditions of acidic/alkaline solution and enzyme hydrolysis [33]. The relative crystallinity of FJI was 35.84%, close to the 34.24% crystallinity index of ginger IDF [12].

### 3.3. Proximate Composition of Bread

The results of the proximate composition parameters of the control and FJI-added bread samples were tabulated in Table 2. There was a concentration-dependent increase in the moisture content of supplemented bread samples. With increases in FJI supplementation of 2 to 8%, a significant upward trend was observed for total DF (2.23–5.10%). The fat content in bread gradually decreased with the addition of FJI, and the control sample exhibited the highest fat content (14.45%). Reducing fat content could extend the shelf life of bread by inhibiting oxidative rancidity during long-term storage [34]. Contrary to fat, protein and ash content increased concentration-dependently in the FJI-supplemented bread. The potential contribution of FJI to enriching the nutritional profile of bread samples can explain the increase in ash content. Compared with the control sample, the carbohydrate content of supplemented samples decreased significantly. In addition, a slight decline in energy value was observed for supplemented samples with increased FJI levels. Generally, intended consumers preferred products with lower energy values owing to their concern for healthy nutrition and weight control [35].

### 3.4. Color Attributes of Bread

A comparison of crust and crumb color properties (*L**, *a**, and *b**.) between control and FJI-supplemented samples was shown in Table 3. *L** reflects the brightness of the bread. The addition of FJI significantly decreased the brightness (*L** value) and redness (*a** value) of the bread crust and crumb. Compared to the control, FJI resulted in a decrease in the yellowness (*b** value) of the bread crust and an increase in the yellowness of the crumb. This was consistent with previous findings that the addition of ginseng-IDF significantly changed the Hunter color values of wheat bread [4]. Color characteristics deepened on the properties of the material. *a** was related to the content of mineral elements, and *b** was related to the content of yellow pigment [36]. Plant pigment suffered thermolysis and/or oxidative degradation during baking, and the extent of degradation decreased toward the center. Increased synthesis of brown-colored melanoidins during baking could also cause a decrease in the degree of lightness and yellowness and an increase in the degree of redness [37]. Consumers prefer darker bread, even if the color is related to the different composition or browning process [14]. In addition, plant additives have the potential to serve as natural food dyes, which are more favorably accepted by consumers compared to the prevalent synthetic or semi-synthetic dyes in the food industry [38].

### 3.5. Specific Volume of Bread

The specific volume of control and FJI-supplemented samples was presented in Table 3. FJI-supplemented bread showed a decrease in specific volume from 3.02 mL/g to 2.45 mL/g with an increase in FJI levels. This result was consistent with the finding of the durum wheat bread supplemented with spinach powder [1]. Loaf volume reduction could be explained by two factors: (1) a fiber-weaned dough or a crippled dough structure made it unable to hold fermented CO_2_, and (2) the fiber competes with gluten proteins for water, negatively affecting gluten hydration and gluten network formation [39]. In addition, the reduction in bread volume might be related to enzyme and yeast activity, while amylase in wheat flour may be inhibited by antioxidant phenolics in FJI, which reduced the level of maltose in the dough, resulting in reduced air production, smaller volume, and denser texture [40].

### 3.6. Texture Properties of Bread

Table 3 showed the influence of FJI on the texture properties of bread. With the addition of FJI, bread hardness significantly increased. The hardness of bread increased by 502.88% with the addition of 8% (*w/w*) of FJI. A similar trend was observed for the chewiness of bread when FJI levels increased. These results may be due to the rigidity of the DF or its competition with the wheat flour component in terms of water absorption, thus strengthening the gluten framework [41]. In addition, the control samples had significantly higher springiness and resilience than the FJI-added bread groups. However, there was no significant difference in the resilience of bread added with 2% FJI, 4% FJI, and 6% FJI. The dense internal structure of bread was formed by combining gelatinized starch granules combined with sugars and lipids [42]. The significantly decreased resilience might be due to a reduced number of air bubbles in the internal framework and matrices of the bread [4]. FJI supplementation above the 2% level resulted in a darker overall appearance and smaller porous structures compared to the control samples. A specific volume indicated the final amount of gas retained in the bread made, significantly impacting consumer preference and acceptance [43]. Increasing fiber particles might impede the development of a good gluten network, inhibiting the viscoelastic network and the weakening of dough, ultimately decreasing the loaf volume during dough formation [44]. Compared to the control, the bread samples with 2% FJI addition had acceptable textural properties.

### 3.7. TPC, TFC, and Antioxidant Activities of Bread

Compared to the control samples (77.47 mg GAE/100 g), the TPC of FJI-addition bread showed a concentration-dependent increase (ranging from 84.08 to 113.01 mg GAE/100 g; Table 3). A similar trend was observed for TFC, where the supplemented samples had significantly higher TFC than that of the control (5.55 mg RE/100 g). Meanwhile, the antioxidant activities in FJI-added wheat bread showed concentration-dependent increases. Compared to the control, 2% FJI supplementation increased108.11%, 30.16%, and 62.02% for ABTS, DPPH, and FRAP, respectively. Significant increases in TPC, TFC, and antioxidant activities were observed due to phenolic compounds like quercetin, catechin, ellagic acid, rutin, gallic acid, syringic acid, eriodictyol, pyrocatechol, syringic, cinnamic acids, and eriocitrin in feijoa [8]. As polyphenolic compounds were associated with high antioxidant potential [45], this might explain the high free radical scavenging activity, TPC, and TFC in the FJI-added bread samples. These results were in agreement with data published previously, in which steamed bread supplemented with sorghum powder [46], black tea [47], and lemon fiber [41] revealed increased TPC, TFC, and antioxidant activities.

### 3.8. Sensory Properties of Bread

The sensory characteristics of a food product determine the acceptance and preference of the consumer and undoubtedly determine the product’s success. In this study, we evaluated five major sensory properties (appearance, flavor, taste, texture, and overall acceptability), and the results were shown in Figure 2. Among all sensory properties, the control sample scored the highest (8.15–8.24) on a nine-point hedonic scale. Adding 2% FJI would not significantly affect the appearance, flavor, taste, and texture of bread. The graininess and coarseness of the bread increased with the corresponding increase in the amount of FJI addition, which greatly affected the taste of the bread. The lower sensory characteristics of the bread with FJI addition compared to the control bread were similar to previous research conducted on various functional ingredients (i.e., onion residue, spinach powder, Doum fruit dietary fiber, and black tea) [1,47,48,49].

### 3.9. In Vitro Adsorption Capacities of Bread

#### 3.9.1. Bile Acid Salt Absorption

The binding capacity for bile acid salt and cholesterol of DF was commonly considered as an essential parameter to assess the binding capacity for the lipophilic component. Bile acid is synthesized in the human liver and subsequently stored within the gall bladder. After food is eaten, bile acid enters small intestine, participates in the circulation of liver sausage, and adjusts the balance of cholesterin [50]. As shown in Figure 3A, the control exhibited the lowest bile acid salt absorption in comparison with all FJI-added breads. Adding FJI can significantly improve absorption capacity, but no obvious difference was observed between the adjacent two FJI-supplemented groups. A concentration-dependent increase in absorptive capacity was observed in ginseng-IDF-supplemented samples [12]. The addition of FJI helped to inhibit cholesterol absorption and promote its excretion, probably due to its chemisorption [51]. A more hydrophobic DF had a higher affinity for sodium cholate than a fiber with a less hydrophobic group [23].

#### 3.9.2. NO^2−^ Absorption Capacity

Assessing the NO^2−^ absorption capacity of IDF from various food sources holds significant value as nitrite is a potential carcinogen [27]. Due to its porous structure, DF can scavenge nitrite ions, leading to decreased absorption of nitrite into tissues and blood, thus hindering the production of carcinogenic N-nitroso compounds in animals [52]. The nitrite adsorption of bread samples was investigated at pH 2.0 and at 7.0 to simulate the gastric and intestinal environment in vitro (Figure 3B). All supplemented samples showed considerably higher adsorption capacity at pH 2.0 than pH 7.0, which was consistent with the results obtained from ginseng-IDF [12]. Compared with all samples supplemented with FJI, the control group showed the lowest absorption of NO^2−^. At pH 2.0, FJI addition levels increased from 2% to 8%, corresponding to an increase in NO^2−^ absorption in a concentration-dependent manner. Comparatively, the difference of NO^2−^ absorption in the case of 4% and 6% FJI-supplemented samples was not so much apparent at pH 7.0. Under acidic conditions, NO^2−^ and H^+^ ions combine to produce HNO_2_, leading to the production of a large number of oxy-nitrogen compounds, including N_2_O_3_, which exhibits a strong electron affinity and may form bonds with negatively charged oxygen molecules of the phenolic acid group of FJI, leading to improved absorption capacity [53,54].

#### 3.9.3. Cholesterol Absorption Capacity

DF had beneficial effects on blood cholesterol attenuation by decreasing the absorption and utilization of over-supplied triglyceride and cholesterol in the small intestine [55]. The addition of FJI significantly enhanced cholesterol absorption capacity at two pH values (Figure 3C). Generally, FJI absorbed cholesterol more efficiently at pH 7.0 than at pH 2.0, which was consistent with the results of DF in foxtail millet bran at pH 2.0 [28]. Thus, cholesterol adsorption of FJI was more likely to occur in the digestive system than in the stomach. Similarly, ginger IDF-supplemented bread had a reasonably good capacity to absorb cholesterol at pH 7.0 compared to pH 2.0 [4]. Our results suggested that the cholesterol adsorption capacity of FJI was obviously affected by pH, which was significantly higher in the simulated intestinal environment (pH = 7.0) than in the simulated gastric environment (pH = 2.0) [56].

### 3.10. Glucose Release Profile of Bread

Previous studies have proved that DF intake from natural foods improves glycemic control in patients with type 2 diabetes mellitus [57]. Postprandial glucose levels can be controlled by IDF consumption due to delayed bioavailability and absorption of glucose [58]. In this study, the glucose release in all samples increased in the first 90 min of digestion and then decreased at 120 min (Table 4). The control samples showed significantly higher glucose release than the FJI-added samples at 0 min, 20 min, 60 min, and 90 min. In addition, no significant difference was observed between the samples supplemented with 4% and 6% at all time intervals. These results suggested that the addition of FJI had a significant hypoglycemic effect on glucose release, which was in line with the findings of Li et al. [59], who reported that supplementation of barley IDF could improve the control of fasting blood glucose.

## 4. Conclusions

This study suggested that FJI had abundant mineral elements, desirable physico-chemical properties, and distinct functional groups of cellulosic polysaccharides. The supplementation of FJI significantly affected the baking value of wheat bread, which contributed to increased antioxidant capacities, while decreasing the carbohydrates and energy value. In addition, FJI-added bread exhibited notable bile acid salt adsorption, a hypoglycemic effect on glucose release, and improved NO^2−^ adsorption at pH 2.0 and cholesterol adsorption at pH 7.0. These results indicated that FJI possessed desirable physiological value and adsorption function, thus positioning it as a promising functional ingredient in food production.

## Figures and Tables

**Figure 1 foods-12-02019-f001:**
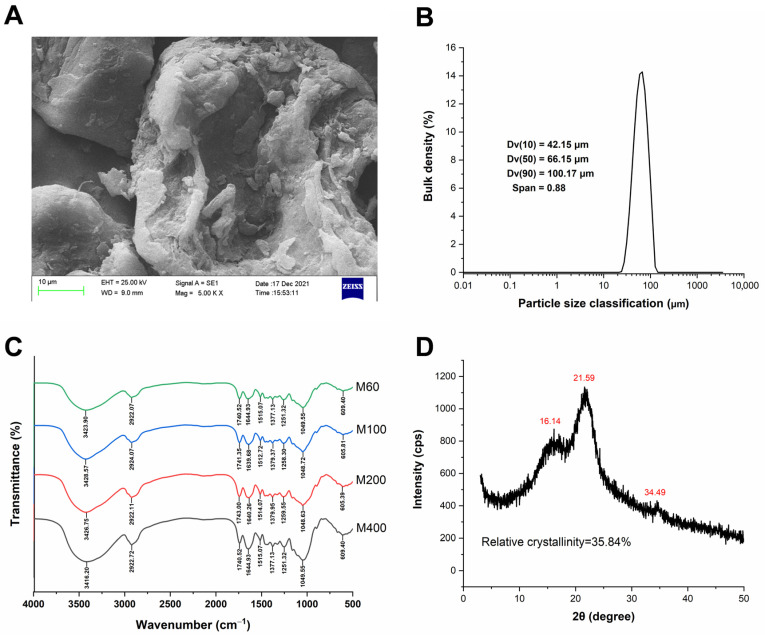
FJI structure. (**A**) SEM image; (**B**) Particle size distribution; (**C**) FT-IR spectra; (**D**) X-ray diffraction spectrum.

**Figure 2 foods-12-02019-f002:**
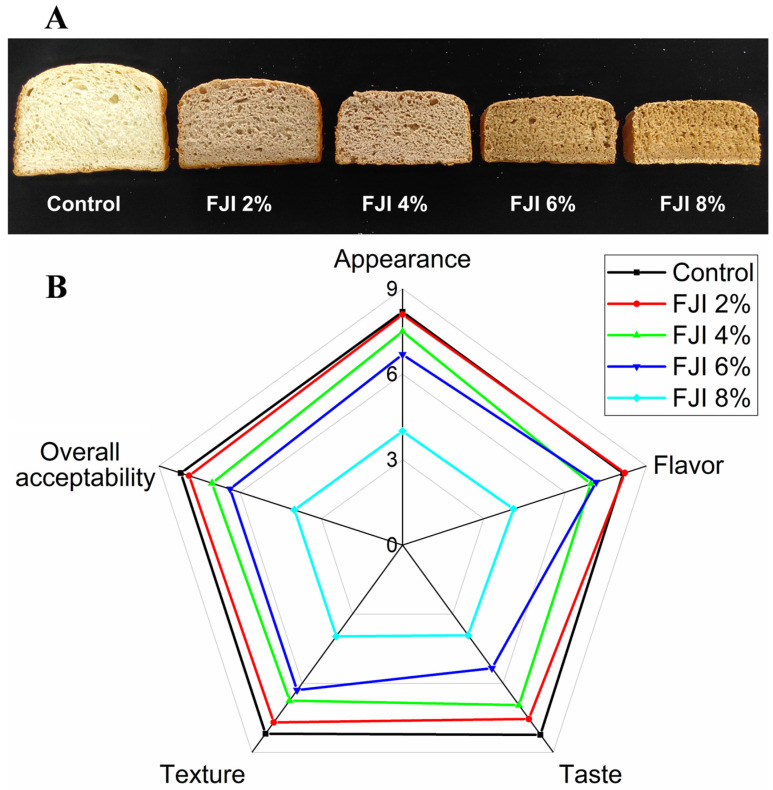
Effect of FJI on physical appearance of wheat bread. (**A**) Images of wheat bread supplemented with FJI; (**B**) The sensory evaluation.

**Figure 3 foods-12-02019-f003:**
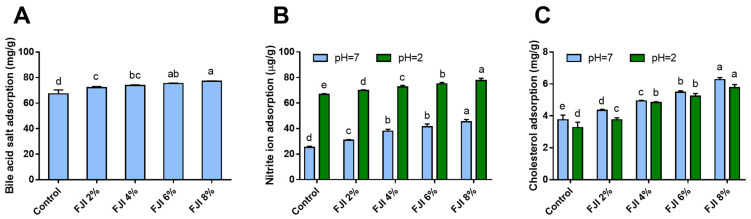
In vitro adsorption capacity of wheat bread supplemented with FJI. (**A**) NO^2−^; (**B**) cholesterol; (**C**) bile acid salt. Different lowercase letters indicate a significant difference (*p* < 0.05).

**Table 1 foods-12-02019-t001:** Mineral composition and physicochemical properties of FJI.

Mineral (Dry Weight)	Value	Physicochemical Properties (Dry Weight)	Value
K(mg/kg)	2847.47 ± 77.20	Water-holding capacity (WHC, g/g)	4.66 ± 0.37
Na(mg/kg)	388.04 ± 22.06	Oil–holding capacity (OHC, g/g)	4.18 ± 0.07
Ca (mg/kg)	321.41 ± 10.76	Swelling capacity (SC, mL/g)	2.09 ± 0.16
Fe (mg/kg)	84.61 ± 3.66	Total phenolic content (TPC, mg GAE/g)	1.63 ± 0.22
Zn(mg/kg)	80.89 ± 2.40	Total flavonoid content (TFC, mg RE/g)	1.15 ± 0.46
Mn(μg/kg)	873.41 ± 79.02	ABTS (μmol TE/g)	3.63 ± 0.17
Cu (μg/kg)	111.48 ± 5.48	DPPH (μmol TE/g)	10.68 ± 0.25
		FRAP (μmol TE/g)	5.40 ± 0.05
		Bile acid salt adsorption (mg/g, pH = 7)	61.69 ± 0.28
		NO^2−^ adsorption (μg/g, pH = 7)	140.33 ± 3.75
		NO^2−^ adsorption (μg/g, pH = 2)	146.31 ± 4.16
		Cholesterol adsorption (mg/g, pH = 7)	14.93 ± 0.22
		Cholesterol adsorption (mg/g, pH = 2)	7.14 ± 0.14

The values represent the mean ± standard deviation of three determinations.

**Table 2 foods-12-02019-t002:** Proximal composition of wheat bread with FJI.

Samples	Moisture (%)	Total DF (%)	Fat (%)	Ash (%)	Protein (%)	Carbohydrate (%)	Energy Value (kJ/100 g)
Control	34.57 ±0.13 b	1.46 ± 0.04 e	14.45 ± 0.02 a	0.83 ± 0.03 c	14.06 ± 0.46 b	36.30 ± 0.26 a	1378.87 ± 5.06 a
FJI 2%	34.95 ± 0.49 b	2.23 ± 0.01 d	14.20 ± 0.01 b	0.92 ± 0.01 b	14.81 ± 0.42 b	35.40 ± 0.66 b	1364.38 ± 10.25 b
FJI 4%	36.25 ± 0.74 a	2.96 ± 0.30 c	13.98 ± 0.05 c	0.95 ± 0.04 b	15.85 ± 0.28 a	33.46 ± 0.26 c	1351.68 ± 4.53 b
FJI 6%	36.57 ± 0.23 a	4.07 ± 0.09 b	13.81 ± 0.05 d	1.03 ± 0.02 a	16.25 ± 0.31 a	32.07 ± 0.64 d	1328.71 ± 9.37 c
FJI 8%	36.67 ± 0.12 a	5.10 ± 0.09 a	13.53 ± 0.06 e	1.07 ± 0.02 a	16.59 ± 0.25 a	31.90 ± 0.25 e	1321.01 ± 4.39 c

The values represent the mean ± standard deviation of three determinations; Control is done without FJI; FJI is Feijoa IDF. Different letters in a column indicate significant differences (*p* < 0.05).

**Table 3 foods-12-02019-t003:** Color parameters, specific volume, texture properties, and antioxidant activities of wheat bread with FJI.

	Control	FJI 2%	FJI 4%	FJI 6%	FJI 8%
Crust color					
*L**	67.91 ± 1.72 a	59.16 ± 1.21 b	54.13 ± 0.42 c	51.32 ± 1.03 d	49.63 ± 0.89 d
*a**	8.89 ± 0.57 c	10.25 ± 0.23 b	10.91 ± 0.57 b	13.07 ± 0.44 a	13.55 ± 0.51 a
*b**	34.26 ± 0.22 a	29.23 ± 0.15 b	28.01 ± 1.59 bc	30.75 ± 0.11 c	27.99 ± 1.09 c
Crumb color					
*L**	68.34 ± 1.92 a	52.43 ± 0.53 b	49.41 ± 0.63 c	47.90 ± 1.21 c	44.53 ± 0.59 d
*a**	1.48 ± 0.17 d	5.84 ± 1.43 c	10.84 ± 0.46 b	11.39 ± 0.17 b	12.81 ± 0.54 a
*b**	15.64 ± 1.27 d	19.38 ± 0.79 c	26.05 ± 0.60 b	26.58 ± 0.37 b	28.87 ± 0.84 a
Specific volume (mL/g)	3.61 ± 0.06 a	3.02 ± 0.07 b	2.76 ± 0.05 c	2.58 ± 0.03 d	2.45 ± 0.02 e
Hardness (g)	68.86 ± 5.83 e	158.99 ± 2.71 d	332.73 ± 22.14 c	367.78 ± 23.86 b	415.15 ± 5.52 a
Chewiness (g)	50.63 ± 6.71 e	92.62 ± 1.89 d	158.06 ± 3.21 c	177.38 ± 3.68 b	191.95 ± 9.97 a
Springiness (%)	96.53 ± 2.44 a	91.30 ± 0.85 b	86.20 ± 0.89 c	81.47 ± 0.15 d	77.87 ± 2.17 e
Resilience (%)	29.00 ± 1.40 a	24.00 ± 1.30 b	23.10 ± 0.30 b	22.70 ± 0.20 b	20.30 ± 0.80 c
TPC (mg GAE/100 g)	77.47 ± 2.53 e	84.08 ± 0.95 d	92.07 ± 1.66 c	99.50 ± 2.48 b	113.01 ± 1.72 a
TFC (mg RE/100 g)	5.55 ± 0.24 d	6.64 ± 0.39 c	7.07 ± 0.05 c	7.78 ± 0.46 b	9.75 ± 0.19 a
ABTS (μmol TE/g)	0.37 ± 0.03 e	0.77 ± 0.08 d	0.95 ± 0.10 c	1.72 ± 0.08 b	2.01 ± 0.05 a
DPPH (μmol TE/g)	0.14 ± 0.02 e	0.40 ± 0.02 d	0.61 ± 0.05 c	0.99 ± 0.02 b	1.28 ± 0.03 a
FRAP (μmol TE/g)	0.10 ± 0.01 e	0.43 ± 0.01 d	0.66 ± 0.06 c	0.93 ± 0.07 b	1.35 ± 0.03 a

The values represent the mean ± standard deviation of three determinations; Control is done without FJI; FJI is Feijoa IDF. Different letters in a row indicate significant differences (*p* < 0.05).

**Table 4 foods-12-02019-t004:** Glucose release content of FJI-enriched breads. (Unit: mg/g).

	0 Min	20 Min	60 Min	90 Min	120 Min
Control	170.63 ± 4.76 a	189.25 ± 1.29 a	205.42 ± 7.87 a	230.78 ± 2.59 a	194.76 ± 5.18 a
FJI 2%	158.13 ± 4.66 b	175.16 ± 3.45 b	188.48 ± 8.27 b	203.63 ± 1.30 b	200.26 ± 3.95 a
FJI 4%	150.47 ± 7.49 bc	158.48 ± 9.46 c	172.94 ± 7.23 c	187.85 ± 3.94 c	184.35 ± 9.79 b
FJI 6%	142.59 ± 4.29 c	154.63 ± 8.48 c	167.30 ± 7.34 c	186.19 ± 1.48 c	179.37 ± 3.44 b
FJI 8%	123.68 ± 3.04 d	140.29 ± 4.74 d	160.62 ± 1.29 c	176.22 ± 6.74 d	175.92 ± 2.73 b

The values represent the mean ± standard deviation of three determinations; Control is done without FJI; FJI is Feijoa IDF. Different letters in a column indicate significant differences (*p* < 0.05).

## Data Availability

Data is shown in the article.

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
