# Peer review of "Effect of Insoluble Dietary Fiber Extracted from Feijoa (Acca sellowiana (O. Berg) Burret.) Supplementation on Physicochemical and Functional Properties of Wheat Bread"

_foods, 2023, doi:10.3390/foods12102019_

Round 1

Reviewer 1 Report

This work presents In vitro adsorption capacities, starch digestibility, and physiochemical properties of wheat bread added with insoluble dietary fiber extracted from feijoa (Acca sellowiana (O. Berg) Burret.) 

Paper format: Ok. As per FOOD format.

Title: Not Fine

Abstract:  The abstract need revision in terms of including significant values among different data. Avoid theoretical approach in abstract, just show what results are saying. Keep to the point and address novelty of the work. The abstract should always be concise and informative. The arguments of why your study is important not making any sense. Extensive revision is required in abstract, as the present sentences sounds noisy during reading. Overall, the abstract is not informative enough and need to show the actual picture of the work.

1.     While the market demand for baked, nutrition-ally fortified, and sensory properties-based products with health benefits has increased sharply (1). This sentence needs proper justification, mean any number or values, how much increased in market?

2.     Feijoa (Acca sellowiana) is a woody species and has been cultivated in some regions of the world, such as New Zealand, USA, France and Australia. Feijoa fruit is sweet/sour 40 and highly aromatic. What about China? No information?

3.     In table 3. The antioxidant activities are too low, I suggest no need to report such poor activities, please rephrase all the discussion upon table. 3

4.     Fig 1. B the particle size is in micro-mete or nano-meter? Mention unit please

Some general comments

Keywords: Please change it to aims of the study, be specific

1. Clarify the research question: The title of the study is quite long and complex, which might make it difficult for readers to quickly grasp what the study is about. I suggest that the authors try to simplify the title and provide a clear research question that outlines the objective of the study.

2. Provide more detailed methods: The study appears to involve in vitro adsorption capacities, starch digestibility, and physiochemical properties, but it's not clear how these were measured or analyzed. I suggest that the authors provide more detailed descriptions of their methods so that readers can better understand how the data was collected and analyzed.

3. Include more context: Feijoa is not a commonly known ingredient, so it might be helpful for the authors to provide more context about its use in food and how it has been studied in the past. This could help readers better understand the significance of the current study.

4. Consider the implications for human health: While the study appears to focus on the properties of bread, it's important to consider the potential implications for human health. I suggest that the authors discuss the potential health benefits of adding insoluble dietary fiber from feijoa to bread, and whether this could have any implications for preventing or managing certain health conditions.

5. Provide clear conclusions: I suggest that the authors provide clear conclusions that summarize their findings and their implications. This will help readers better understand the significance of the study and its potential impact.

Author Response

Dear Reviewer:

Thank you very much for the consideration of our manuscript to be published in the prestigious Journal of Foods. I accept and appreciate your comments, which are all valuable and very helpful for revising and improving our paper, as well as the important guiding significance to our research. We have studied the comments carefully and have made a correction which we hope meets with approval. The revised portion is marked in red in the manuscript. We also have checked our revised manuscript for proper English language, grammar, punctuation, spelling, and overall style by a professional. The main corrections in the manuscript and the response to your comments are as flowing:

Point 1. Abstract:  The abstract need revision in terms of including significant values among different data. Avoid theoretical approach in abstract, just show what results are saying. Keep to the point and address novelty of the work. The abstract should always be concise and informative. The arguments of why your study is important not making any sense. Extensive revision is required in abstract, as the present sentences sounds noisy during reading. Overall, the abstract is not informative enough and need to show the actual picture of the work.

Response 1: We are deeply grateful for these constructive comments. We have revised the abstract as your suggested.

Point 2: While the market demand for baked, nutrition-ally fortified, and sensory properties-based products with health benefits has increased sharply (1). This sentence needs proper justification, mean any number or values, how much increased in market?

Response 2: Thanks for your suggestion. We did not find any relevant data about how much increased in market. We changed the sentence into ‘Foods having high protein and fiber content are now generally preferred by consumers to maintain their health and to act against many diseases such as diabetes and obesity. As a result, a contemporary market trend is emerging with the development of products that offer both health benefits and desirable sensory attributes’.

Point 3: Feijoa (Acca sellowiana) is a woody species and has been cultivated in some regions of the world, such as New Zealand, USA, France and Australia. Feijoa fruit is sweet/sour and highly aromatic. What about China? No information?

Response 3: According to Zhu (2018), the production quantity of feijoa in New Zealand has reached over 800 tons and other countries such as China is also developing the fruit for cultivation. Researchers at Shanghai Botanical Garden first brought F. sellowiana to China from Europe in the 1980s (Han et al., 2009). Thereafter, it was introduced to Jiangsu, Sichuang, and Hunan provinces. We have added this information in our revised manuscript.

Point 4: In table 3. The antioxidant activities are too low, I suggest no need to report such poor activities, please rephrase all the discussion upon table. 3

Response 4: Many thanks for the comments and remind. Previous study has reported that TPC in wheat flour with yerba mate was 82.95 mg GAE/100 g, ABTS, and FRAP were 443.80 and 95.84μmoLTE/g, respectively (Santetti et al., Journal of Food Science, 2021). In addition, recent years have seen a profusion of studies on bread supplemented with various phenolic compounds, such as phenolic acids, green tea extracts, black tea powder and grape seed (Xu et al., Journal of Functional Foods, 2019). Compared to the bread with these added ingredients, the antioxidant activity of the FJI-added bread was indeed lower. This might due to the different food resources and substances. Compared to the same additional substance (IDF), DPPH, ABTS and FRAP in 8% Ginseng IDF-supplemented bread samples were 1.68, 1.89 and 2.08 μmoL TE/g, respectively. In our study, they were 1.28, 2.01 and 1.35 μmoL TE/g. The antioxidant activity of these IDF added breads did not differ much. Although the antioxidant activity of IDF-added bread was relatively low, the data could provide some important information and reference for the utilization of IDF on food.

Point 5: Fig 1. B the particle size is in micro-mete or nano-meter? Mention unit please

Response 5: Many thanks for the comments and remind. We have added the particle size (μm) in our revised manuscript.

Some general comments

Point 6: Keywords: Please change it to aims of the study, be specific

Clarify the research question: The title of the study is quite long and complex, which might make it difficult for readers to quickly grasp what the study is about. I suggest that the authors try to simplify the title and provide a clear research question that outlines the objective of the study.

Response 6: We really appreciate your kind help with detailed guidance. We have changed the keywords ‘in vitro adsorption capacities’ into ‘functional properties’, and changed the tittle into ‘Effect of insoluble dietary fiber extracted from feijoa (Acca sellowiana (O. Berg) Burret.) Supplementations on Physicochemical and Functional Properties of Wheat Bread’.

Point 7: Provide more detailed methods: The study appears to involve in vitro adsorption capacities, starch digestibility, and physiochemical properties, but it's not clear how these were measured or analyzed. I suggest that the authors provide more detailed descriptions of their methods so that readers can better understand how the data was collected and analyzed.

Response 7: Thank you for your thoughtful suggestion. We have added more details in the methods in our revised manuscript.

Point 8: Include more context: Feijoa is not a commonly known ingredient, so it might be helpful for the authors to provide more context about its use in food and how it has been studied in the past. This could help readers better understand the significance of the current study.

Response 8: Thank you very much for your constructive suggestion. Survey of the market showed that a range of feijoa derived products has been developed. They include ice cream, chocolate, candy, smoothie, wine, bread spread, jam, yogurt, muffin, puree, juice, and so on. Feijoa based ingredients have potential to be developed for functional food formulation. We have added more details in our manuscript.

Point 9: Consider the implications for human health: While the study appears to focus on the properties of bread, it's important to consider the potential implications for human health. I suggest that the authors discuss the potential health benefits of adding insoluble dietary fiber from feijoa to bread, and whether this could have any implications for preventing or managing certain health conditions.

Response 9: Thank you very much for your constructive suggestion. We have added more information about the potential health benefits of feijoa IDF (In introduction and discussion section).

Point 10: Provide clear conclusions: I suggest that the authors provide clear conclusions that summarize their findings and their implications. This will help readers better understand the significance of the study and its potential impact.

Response 10: Thanks for your careful advice. We have revised our conclusion as you suggested.

Reviewer 2 Report

Introduction

line 54 - a new paragraph should be created here

The introduction, especially in the first paragraph, should be developed using the suggested literature: https://doi.org/10.3390/antiox11112178

Materials and Methods

line 63- The authors should complete the information regarding the ingredients used to bake the bread:“High-gluten flour” - from what grain was it made and what level of gluten did the flour have?; How much % fat did the butter have?; “non-fat dry milk” - was it cow milk?

In titles of subsections 2.4 and 2.5 please add that they refers to Feijoa IDF

2.5. Determination of mineral content - Please state from where the methodology was chosen, and detail the parameters for measuring elemental content by FAAS and GFAAS.

2.6. Bread-making process - how many loaves of each variant were produced? please provide this information in this subsection

line 124- white flour but made from what?; hot dough? - but how was made and based on what ingredients?

line 126- “FJI was added at 0.0, 4.0, 8.0, 12.0, and 16.0 g/200 g flour” - which component was appropriately reduced when Feijoa IDF was added?

line 140-142- energy value of bread was calculated in kJ/100g or kcal/100g ? Please verify because you wrote these two units. In calculating caloric content, the authors omitted the content and caloric content of fiber, which was certainly present in the bread. Please correct the formula and correct the values from the calorie calculation.

2.8. Determination of the specific volume - How was the volume of the loaves determined?

2.9. Color analysis - please provide more information on the method of color determination; state that the color of the crust and crumb of bread samples was analyzed

2.16. Statistical analysis- What test was chosen to determine homogeneous groups? please complete

Results and discussion

The presentation of the results in tables and graphs is clear and informative. All tables and graphs are needed and do not repeat information.

The authors referred to all the presented research results. In each subsection, they discussed the presented results in a logical way based on previously published articles by themselves and other researchers.

The authors should use literature preferably from the last few years. I suggest enriching the literature used in the discussion of the results with two items: https://doi.org/10.3390/molecules26247564; https://doi.org/10.3390/molecules26154641.

Conclusions

line 419-421- this sentence should be removed because it is a repetition of information already given and these are not conclusions

The rest of the text presents a good summary of the collected data and contains key conclusions for the study. The conclusions follow from the authors' work and are original.

Author Response

Dear Reviewer:

Thank you very much for the consideration of our manuscript to be published in the prestigious Journal of Foods. I accept and appreciate your comments, which are all valuable and very helpful for revising and improving our paper, as well as the important guiding significance to our research. We have studied the comments carefully and have made a correction which we hope meets with approval. The revised portion is marked in red in the manuscript. We also have checked our revised manuscript for proper English language, grammar, punctuation, spelling, and overall style by a professional. The main corrections in the manuscript and the response to your comments are as flowing:

Point 1: Introduction

line 54 - a new paragraph should be created here

The introduction, especially in the first paragraph, should be developed using the suggested literature: https://doi.org/10.3390/antiox11112178; 

Response 1: Thanks for your careful advice. We have revised our introduction to help readers better understand the background of the study.

Point 2: Materials and Methods

line 63- The authors should complete the information regarding the ingredients used to bake the bread:“High-gluten flour” - from what grain was it made and what level of gluten did the flour have?; How much % fat did the butter have?; “non-fat dry milk” - was it cow milk?

In titles of subsections 2.4 and 2.5 please add that they refer to Feijoa IDF

Response 2: Thanks for your careful advice. In this study, we have added the information about high-gluten wheat flour (protein content 12.2g/100g), the butter had 81.0g/100g fat, and ‘non-fat dry milk power’ in our study was produced from fresh cow's milk which is evaporated and concentrated and spray-dried. In addition, we have revised the tittle of subsections 2.4 and 2.5.

Point 3: 2.5. Determination of mineral content - Please state from where the methodology was chosen, and detail the parameters for measuring elemental content by FAAS and GFAAS.

Response 3: Thanks for your careful advice. We have cited relevant literature and added details in our revised manuscript.

Point 4: 2.6. Bread-making process - how many loaves of each variant were produced? please provide this information in this subsection

Response 4: Each recipe can make three loaves. We have added this information in our revised manuscript.

Point 5: line 124- white flour but made from what?; hot dough? - but how was made and based on what ingredients?

Response 5: In this study, The bread was prepared using the scalding process and secondary fermentation method. Firstly, 120 g boiling water was mixed with 100 g wheat flour (protein content 12.2g/100g) to create hot dough. The bread formulation included 194 g of wheat flour, 12.5 g of hot dough, 116 g of water, 8.5 g of non-fat dry milk, 30.0 g of butter (fat content 81.0g/100g), 25.0 g of sugar, 3.0 g of dry yeast and 2.50 g of salt. Wheat flour was replaced with FJI at 0.0, 4.0, 8.0, 16.0 g/200 g flour.

Point 6: line 126- “FJI was added at 0.0, 4.0, 8.0, 12.0, and 16.0 g/200 g flour” - which component was appropriately reduced when Feijoa IDF was added?

Response 6: Wheat flour was replaced with FJI by the weight of the flour. We have check and corrected this sentence.

Point 7: line 140-142- energy value of bread was calculated in kJ/100g or kcal/100g ? Please verify because you wrote these two units. In calculating caloric content, the authors omitted the content and caloric content of fiber, which was certainly present in the bread. Please correct the formula and correct the values from the calorie calculation.

Response 7: Thanks for your careful suggestion. It was a spelling error. We have check and corrected this formula.

Point 8: 2.8. Determination of the specific volume - How was the volume of the loaves determined?

Response 8: The loaf volume was evaluated using a standard millet displacement method. We have added this detail in our revised manuscript.

Point 9: 2.9. Color analysis - please provide more information on the method of color determination; state that the color of the crust and crumb of bread samples was analyzed

Response 9: Crust color was evaluated at 3 points on each surface of bread. Crumb color was determined at the middle point of the central 2 cm thick slice. L* denotes the color brightness, a* red/green saturation, and b* yellow/blue saturation.

Point 10: 2.16. Statistical analysis- What test was chosen to determine homogeneous groups? please complete

Response 10: Statistical analyses of results obtained were conducted with the one-way analysis of variance (ANOVA). Significant differences (p ≤ 0.05) between the mean values were determined using the Duncan’s Multiple Range Test. We have added this detail in our revised manuscript.

Point 11: Results and discussion

The presentation of the results in tables and graphs is clear and informative. All tables and graphs are needed and do not repeat information. The authors referred to all the presented research results. In each subsection, they discussed the presented results in a logical way based on previously published articles by themselves and other researchers. The authors should use literature preferably from the last few years. I suggest enriching the literature used in the discussion of the results with two items: https://doi.org/10.3390/molecules26247564; https://doi.org/10.3390/molecules26154641.

Response 11: Thanks for your careful advice. We have added these two related reference in our revised manuscript.

Point 12: Conclusions

line 419-421- this sentence should be removed because it is a repetition of information already given and these are not conclusions

The rest of the text presents a good summary of the collected data and contains key conclusions for the study. The conclusions follow from the authors' work and are original.

Response 12: Thanks for your valuable suggestion. We have revised the conclusion in our revised manuscript.

Reviewer 3 Report

I have read the manuscript and believe it is well prepared for publication. Compared to similar studies, it is distinguished by a wide and justified range of research methods used. Authors should carefully re-read the text and correct a few minor errors, such as:
- "Aoac" page 3,
- "Mineral dry" table 1,
- FT-IR is not microstructure - figure 1,
- I suggest to divide table 3 into two - treat physical properties (colour, texture) separately, and bioactive ingredients and antioxidant activity separately,
- Conclusions: "according to sensory properties, wheat bread [...] could improve nutritional status..." - I would reconsider it; besides, the addition of DF cannot, by definition, improve nutritional value.

The language quality is satisfactory.

Author Response

Dear Reviewer:

Thank you very much for the consideration of our manuscript to be published in the prestigious Journal of Foods. I accept and appreciate your comments, which are all valuable and very helpful for revising and improving our paper, as well as the important guiding significance to our researches. We have studied comments carefully and have made a correction which we hope meet with approval. Revised portion is marked in red in the manuscript. We also have checked our revised manuscript for proper English language, grammar, punctuation, spelling, and overall style by a professional. The main corrections in the manuscript and the response to your comments are as flowing:

Point 1: - "Aoac" page 3,
- "Mineral dry" table 1,
- FT-IR is not microstructure - figure 1,

Response 1:Many thanks for your careful suggestion. We have changed these minor errors as you suggest and checked throughout this manuscript.

Point 2:- I suggest to divide table 3 into two - treat physical properties (colour, texture) separately, and bioactive ingredients and antioxidant activity separately.

Response 2:Thanks for this careful and professional guideline. Unfortunately, according to another reviewer’s suggestion, there is no need to mention the bioactive ingredients and antioxidant activity, because the antioxidant activities are too low. After careful thinking and weighing, we have decided to maintain the original state and not to emphasize it separately. Thank you again for your nice suggestion.     

Point 3: - Conclusions: "according to sensory properties, wheat bread [...] could improve nutritional status..." - I would reconsider it; besides, the addition of DF cannot, by definition, improve nutritional value.
Response 3:Thanks for your professional suggestion. We have changed ‘nutritional quality’ to ‘functional quality’ and revised the conclusion.

Round 2

Reviewer 1 Report

All the suggestions has been endorsed. Accept in current format

Reviewer 2 Report

The authors have addressed all comments and feedback. They have improved every aspect of the article from the introduction to the methodology and discussion of the results. Also the conclusions have been rewritten. I have no further comments on the article.